# ALDH3A1 Overexpression in Melanoma and Lung Tumors Drives Cancer Stem Cell Expansion, Impairing Immune Surveillance through Enhanced PD-L1 Output

**DOI:** 10.3390/cancers11121963

**Published:** 2019-12-06

**Authors:** Erika Terzuoli, Cristiana Bellan, Sara Aversa, Valerio Ciccone, Lucia Morbidelli, Antonio Giachetti, Sandra Donnini, Marina Ziche

**Affiliations:** 1Department of Medicine, Surgery and Neuroscience, University of Siena, 53100 Siena, Italy; terzuoli8@unisi.it; 2Department of Medical Biotechnology, University of Siena, 53100 Siena, Italy; cristiana.bellan@unisi.it (C.B.); sara.aversas@gmail.com (S.A.); 3Department of Life Sciences, University of Siena, 53100 Siena, Italy; ciccone3@student.unisi.it (V.C.); lucia.morbidelli@unisi.it (L.M.); okkamm@gmail.com (A.G.)

**Keywords:** ALDH3A1, redox metabolism, immune surveillance, stemness, inflammatory mediators, EMT

## Abstract

Melanoma and non-small-cell lung carcinoma (NSCLC) cell lines are characterized by an intrinsic population of cancer stem-like cells (CSC), and high expression of detoxifying isozymes, the aldehyde dehydrogenases (ALDHs), regulating the redox state. In this study, using melanoma and NSCLC cells, we demonstrate that ALDH3A1 isozyme overexpression and activity is closely associated with a highly aggressive mesenchymal and immunosuppressive profile. The contribution of ALDH3A1 to the stemness and immunogenic status of melanoma and NSCLC cells was evaluated by their ability to grow in 3D forming tumorspheres, and by the expression of markers for stemness, epithelial to mesenchymal transition (EMT), and inflammation. Furthermore, in specimens from melanoma and NSCLC patients, we investigated the expression of ALDH3A1, PD-L1, and cyclooxygenase-2 (COX-2) by immunohistochemistry. We show that cells engineered to overexpress the ALDH3A1 enzyme enriched the CSCs population in melanoma and NSCLC cultures, changing their transcriptome. In fact, we found increased expression of EMT markers, such as vimentin, fibronectin, and Zeb1, and of pro-inflammatory and immunosuppressive mediators, such as NFkB, prostaglandin E2, and interleukin-6 and -13. ALDH3A1 overexpression enhanced PD-L1 output in tumor cells and resulted in reduced proliferation of peripheral blood mononuclear cells when co-cultured with tumor cells. Furthermore, in tumor specimens from melanoma and NSCLC patients, ALDH3A1 expression was invariably correlated with PD-L1 and the pro-inflammatory marker COX-2. These findings link ALDH3A1 expression to tumor stemness, EMT and PD-L1 expression, and suggest that aldehyde detoxification is a redox metabolic pathway that tunes the immunological output of tumors.

## 1. Introduction

Aldehyde dehydrogenases (ALDHs), NADP-dependent metabolic enzymes belonging to a large family of molecules, have a diverse tissue- and organ-specific expression pattern and a different cellular subcompartment localization [1,2,3]. ALDHs display a broad spectrum of biological activities covering the control of cellular homeostasis and stemness, and detoxification of toxic endogenous/exogenous products such as alcohol and aldehydes, particularly aldehyde 4-hydroxynonenal (HNE). By increasing NADPH availability during aldehyde detoxification, these enzymes link energetics and metabolism to the redox status of the cells, which in turn uses oxidant signals through transcription factor systems, including nuclear factor-kB (NFkB), to integrate energy efficiency, the resistance of cells to injury, and tissue repair [4]. Some ALDHs, particularly ALDH1A1, ALDH1A3, and ALDH3A1, are widely regarded as markers of stemness in normal tissues (basal skin, bronchial mucosa cells, human cornea, breast, liver), as well as in several solid tumors (non-small-cell lung cancer (NSCLC), melanoma, gastric, breast) [5,6,7,8,9]. However, recent evidence shows that these isozymes, rather than being a mere tag, are distinctly functional in the process of stem cell formation [10,11,12]. Of relevance to this work, the ALDH3A1 isozyme plays an important role in the survival, metastasis, and drug resistance of cancer cells, being mechanistically involved in cancer stem cell expansion and differentiation [6,7]. The significant changes imparted by ALDHs expression on tumor cells (expansion, protein synthesis, differentiation), are likely to impact on the cancer immunity cycle, the multistep process that governs cancer growth dynamics [13]. CSCs escape the immune system by producing molecules that attenuate the immune system, and/or by upregulating inhibitory molecules for the immune cells, such as programmed cell death protein 1 ligands PD-L1 and PD-L2 [14,15,16].

In light of the emerging complexities relating to ALDHs’ influence on cancer cells, we decided to reinvestigate the process of stem cell formation, seeking for a link between stem cells and cancer immune responses. Here, we focused on ALDH3A1 in lung cancer cell lines and melanoma cells, mainly for the availability of patient specimens for these tumors at our institution, the University Hospital of Siena. We demonstrate that ALDH3A1 activity and expression in melanoma and NSCLC cells regulate their stem cell and mesenchymal phenotype, which are closely associated with stemness and immune escape, modulating the levels of PD-L1 expression, reprogramming the tumor microenvironment through the release of immunosuppressive and pro-inflammatory mediators, and the inhibition of T cell proliferation.

## 2. Results

### 2.1. The Expression and Activity of ALDH3A1 in Melanoma and NSCLC Cells

To investigate whether ALDH3A1 induces stemness and influences the expression of molecules involved in tumor immunity, we employed the following tumor cell lines: a) WM266-4, metastatic melanoma cells BRAF wild type, b) HCC4006, metastatic NSCLC cells, c) MEL4478D, cells from a patient with melanoma BRAF wild type (gently provided by Prof. Michele Maio, University of Siena). WM266-4, HCC4006, and MEL4478D are herein termed WM, HCC, and MEL. ALDH3A1 protein and mRNA expression were detected at varying levels in all the above mentioned tumor cells, while they were negligible in normal keratinocytes HaCaT (Figure 1a,b). The enzyme activity was 2–3-fold higher in HCC and MEL than in WM cells, which exhibited lower NADPH production (2.4 ± 0.3 nmol/min/mg protein of NADPH production; Figure 1c). 

Once the experimental system was validated, we created cellular models either overexpressing ALDH3A1 in WM cells (3A1^high^), or silencing ALDH3A1 in WM and HCC cells (3A1^low^ clones Sh #1 and #2) compared to cells with an empty vector (Ctr) (Appendix A). On the basis of the cell transfection, for WM we used the follow terminology: Ctr, 3A1^low^ and 3A1^high^; for HCC and MEL we used this terminology: Ctr (with constitutive high expression and activity of ALDH3A1) and 3A1^low^.

Stable ALDH3A1 silencing produced a decline of enzyme activity (Appendix A), resulting in reduced cell viability and clonogenicity (Appendix A).

### 2.2. ALDH3A1 Activity Elicits a Stem-Cell-Like Phenotype in WM266-4 and HCC4006 Cells

Reportedly, ALDH3A1 plays an important role in stemness in several solid cancers [6,7,8,9]. We examined ALDH3A1’s ability to induce stem-cell-like features in WM and HCC cells by evaluating tumorsphere formation under non adherent culture conditions, and the expression of stem-like markers (Figure 2). In WM and HCC, maintained in a complete medium with 10% FBS, we observed a steady increase in ALDH3A1 protein expression in second- and third-generation spheres, compared to the first- (Figure 2a). Notably, 3A1^high^ exhibited a greater number and size of third-generation spheres (Figure 2b). In contrast, 3A1^low^ cells yielded poorly aggregated third-generation spheres, displaying a significant decrease in spheres (number and size) when compared to control cells (Figure 2b). Furthermore, ALDH3A1 modulation induced a significant perturbation of redox proteome and genes, evaluated in third-generation spheres with the higher expression of the enzyme: low ALDH3A1 triggered the accumulation of 4-HNE-induced covalent adducts in tumorspheres (Figure 2c,f) and a sharp reduction of stem-like markers expression such as Oct4, KLF4, Sox2, Nanog, Twist, and CD133 (Figure 2d–h). In contrast, these markers were increased up to 4-fold in spheres from WM 3A1^high^ with reduced 4-HNE protein adducts (Figure 2d,e). The marked divergence in mRNA values between WM 3A1^high^ and WM 3A1^low^, predominant for Sox2, provided the clue to exploring epithelial mesenchymal transition (EMT) in our cells given its role in epithelial differentiation [17].

### 2.3. Epithelial Mesenchymal Transition (EMT) in Tumor Cells Is Associated with ALDH3A1 Expression

EMT defines the loss of epithelial traits in epithelial cells (loss of e-cadherin, encoded by CDH1, expression). Coupled with the acquisition of mesenchymal characteristics (increase of fibronectin, encoded by FN1, vimentin, encoded by VIM, and Zeb1 encoded by Zeb1 expression), it reduced intercellular adhesion and increased cell motility as well [18]. Reportedly, the EMT process is closely associated with CSCs generation [19]. To investigate whether ALDH3A1 expression might be involved in mesenchymal phenotype development, we studied EMT markers (CDH1, Zeb1, VIM, and FN1) at the mRNA expression level in all stem-cell-like tumor cells (Figure 3a–c). We found a significant overexpression of Zeb1, VIM, and FN1 in 3A1^high^, contrasting with their downregulation in 3A1^low^ cells (Figure 3a–c). Conversely, we observed a CDH1 downregulation in 3A1^high^, differing again from its overexpression in 3A1^low^ cells (Figure 3a–c). By using the Boyden chamber, we assessed the metastatic potential of tumor cells. The test has been performed in the presence of serum, an unspecific chemoattractant agent. After 18 h of incubation, in both cell lines, we detected an important reduction of cells migrated for 3A1^low^ (Figure 3d,e).

Taken together, these pieces of evidence indicate a close relationship between ALDH3A1 expression and tumor EMT development and invasion.

### 2.4. ALDH3A1 Affects Inflammatory Modulators of Immune Surveillance

In light of the changes of stem-cell-like/mesenchymal features elicited by varying the intracellular ALDH3A1 levels, we explored whether the enzyme would affect inflammatory pathways known to amplify tumor progression through various mechanisms, including immune escape [20,21,22]. We found that ALDH3A1 expression in 3A1^high^ cells was associated with the overexpression of COX-2 and mPGES1 (Appendix A), doubling the release of PGE-2 (Figure 4a and Appendix A). Conversely, in 3A1^low^ cells (Figure 4b and Appendix A), PGE-2 output was dramatically reduced. Similar results were obtained in MEL (Figure 4a, last column, and Appendix A). Of note, ALDH3A1 expression was also associated with NFkB overexpression and activity, measured as the nuclear localization of p65 protein by immunofluorescence (Figure 4d). In agreement with literature data demonstrating that PGE-2 by tumor cells influences the expression and release of immunosuppressive cytokines [21], we found that ALDH3A1 silencing selectively reduced immunosuppressive cytokines (Figure 4c), and increased the production of immune-stimulating ones (Figure 4c and Appendix A). These findings were corroborated by mRNA cytokines expression (Appendix A) and the release of the immunosuppressive IL-6 and -13 and immunostimulatory IL-12/23 and IFNγ (Figure 5). Indeed, the release of IL-6 and -13 in 3A1^high^ cells (Ctr for HCC) increased by approximately 50%, while that of IFNγ and IL-12/23 halved (Figure 5a–d). Conversely, we found increased IFNγ and IL-12/23 release in 3A1^low^ (Figure 5a–d). The difference in cytokine release in 3A1^low^ versus 3A1^high^ (or Ctr) was similar in NSCLC and in melanoma cells. The release of cytokines in MEL was similar to that of WM 3A1^high^ (Figure 5e,f). In sum, the NFkB expression and the PGE-2 and cytokine release pattern suggest a close relationship between the extent of redox metabolism, markedly regulated by ALDH3A1 expression in tumor cells, and the formation of an inflammatory and immunosuppressive milieu.

### 2.5. ALDH3A1 Controls PD-L1 Expression in Tumor Cells

Expression of PD-L1 in tumors shields cancer cells from immune-mediated cell death [23,24]. Based on the immune profile acquired by the tumor cells in relation to the ALDH3A1, we evaluated whether PD-L1 levels were also a function of ALDH3A1 expression/activity. We observed the highest PD-L1 expression in WM 3A1^high^ compared to 3A1^low^, and the lowest PD-L1 expression in 3A1^low^ HCC cells versus Ctr (Figure 6a,b). Because of the strong dependence of Zeb1, known to link the EMT regulatory program to PD-L1 expression [25], on ALDH3A1 expression levels (Figure 3), we investigated whether other transcription factors might be involved in PD-L1 expression downstream of ALDH3A1 activity. cMYC and SOX2, two immunomodulatory proteins known to control PD-L1 expression [17,26,27], were found to be overexpressed in 3A1^high^ cells (Ctr for HCC), and downregulated in 3A1^low^ cells (Figure 6a–d), suggesting that ALDH3A1 expression and activity start a complex process involving EMT, stemness, and cytokine release, to contribute to the tumor’s intrinsic immunity. Of note, similar to ALDH3A1, in human melanoma ALDH1A1 was shown to associate with cancer stem-like features and therapy resistance [2]. The ALDH1A1 isozyme oxidizes retinaldehyde to retinoic acid (RA). RA regulates the expression of a variety of genes through RAR and RXR nuclear receptors, which control the transcription of target genes possessing RA-responsive elements (RAREs) [2]. In WM cells, the enzymatic inhibition of ALDH1A1 did not affect cell survival and clonogenicity (Appendix A), and the ALDH1A1 expression did not increase in third-generation tumorspheres, notwithstanding the fact that its pharmacological inhibition decreased the tumorsphere area (Appendix A). ALDH1A1 inhibition had a weak effect on the production of immune-stimulating cytokines (IFNγ: 1.72 and IL-12: 1.34 fold change vs. control cells, Appendix A) and, in particular, the blocking of RAR and RXR nuclear receptors failed to decrease PD-L1 expression in 3A1^high^ cells (Appendix A), indicating the existence of a selective link between ALDH3A1-dependent redox metabolism and intrinsic immunity in our tumor models.

### 2.6. Tumor Intrinsic High ALDH3A1 Activity Reduces PBMC Proliferation

To functionally investigate the effects exerted by cell intrinsic ALDH3A1 on the tumor microenvironment, we set up experiments with tagged peripheral blood mononuclear cells (PBMCs) cultured either in conditioned media (CM) derived from tumor cells or co-cultured with irradiated tumor cells. We monitored the PBMC proliferation by measuring fluorescence peaks up to 144 h. The PBMCs co-cultured with WM and HCC Ctr and 3A1^low^ exhibited a greater proliferative ability (2.6-fold) when incubated with WM 3A1^low^ cells versus 3A1^high^ (Ctr for HCC) cells (Figure 7a–d and Table 1 at 72 h). In order to investigate the impact of cytokines released from 3A1^low^ or 3A1^high^ (Ctr for HCC) cells on the tumor microenvironment, we co-cultured PBMCs with conditioned media (CM) of tumor cells in the presence of anti-CD3/CD28. CM from 3A1^low^ cells increased PBMC proliferation (1.7-fold), while that derived from 3A1^high^ reduced it (Appendix A). This indicates that intrinsic ALDH3A1 activity might drive tumor cells to disable the immunological state of the tumor.

### 2.7. The Expression of ALDH3A1 Correlates with PD-L1 and COX-2 in Melanoma and NSCLC Tumor Samples

To further assess the translational relevance of melanoma and the NSCLC cell-intrinsic ALDH3A1 pathway, we performed ALDH3A1 staining and quantitatively assessed melanoma and NSCLC ALDH3A1 positivity in relation to PD-L1 expression (as markers of oncogenic and immune signals), and COX-2 expression (as marker of tumor-related inflammation) in tumor biopsies obtained from *n* = 7 melanoma patients and *n* = 13 NSCLC patients (Figure 8 and Appendix A). Twelve of 13 lung carcinoma cases were limited to the lung (pT1) and were moderately differentiated (grade G2); one case was non-organ-confined (pT3), but moderately differentiated (grade G2). For lung cancer, patients’ median age was 72 years (range 51–85 years) and there were three females and 10 males. For melanoma cases, three cases were localized tumors (pT1–pT2b) and four cases were regionally spread and metastatic (pT3–pT4b). For melanoma, patients’ median age was 70 years (range 27–86 years) and there were three females and four males. Clinical–pathological characteristics are reported in Appendix A. ALDH3A1 staining was microscopically recognizable as brown cytoplasmic staining. Using the H score, we found that 12 out of 13 (92%) lung adenocarcinoma cases and all seven (100%) melanoma cases were positive for ALDH3A1. Other cellular components of normal lung tissue exhibited strong immunohistochemical expression of ALDH3A1 protein in their cytoplasm; however, type I and II pneumocytes showed faint to no ALDH3A1 expression. Similarly, in skin melanoma samples, normal keratinocytes showed a faint cytoplasmic ALDH3A1 expression. Consistent with our in vitro results, tissues from either lung cancer or melanoma with a high ALDH3A1 H score (i.e., ≥ 4) showed PD-L1 expression higher than 70%, while samples with low ALDH3A1 expression (i.e., ≤ 3) showed low PD-L1 positivity (5–40%) (Appendix A). Additionally, when we analyzed the COX-2 expression in relation to ALDH3A1, we found 11 positive cases out of 13 (84.6%) for lung adenocarcinoma and six positive cases out of seven (85.7%) for skin melanoma, with high H scores for both ALDH3A1 and COX-2 (Figure 8 and Appendix A). These findings suggest a relationship between ALDH3A1 and tumor immunity, thereby indicating the potential translational relevance of cell-intrinsic ALDH3A1 expression/activity in response to PD-1/PD-L1 blockade treatment.

## 3. Discussion

To illustrate the purpose of this work, we first discuss the results of the immunohistochemical analysis of ALDH3A1 expression performed on patient specimens taken from lung adenocarcinoma (NSCLC) and skin melanoma (13 patients for NSCLC and seven for melanoma). 

By comparing the expression levels of ALDH3A1 in patient specimens from both tumors with those of COX-2 and PD-L1, an oncogenic/inflammatory marker and an immune signal, we found that high ALDH3A1 expression was invariably associated with high expression of both COX-2 and PD-L1, in sharp contrast to the consistent downregulation of these markers observed in tumor specimens displaying low expression of the enzyme (Figure 8). These findings suggest that the oncogenic potential of tumor cells is controlled by ALDH3A1 abundance. In addition, these findings have an instructive value in the design of experiments aimed at revealing the mechanism of the enzyme action on tumor cells. To investigate how the varying ALDH3A1 expression levels might affect the carcinogenicity state, we opted to employ tumor cell lines, i.e., NSCLC HCC4006, and melanoma cells WM266-4 and MEL4478D in gain or loss of function studies.

In HCC4006, WM266-4, and MEL4478D cells, high expression of ALDH3A1 revealed incipient stem-cell-like properties, as reflected by the markedly enhanced capability to form tumorspheres in vitro (size and numbers), and by the expression of canonical stem cell markers (Oct4, Sox2, KLF4, and Nanog). Moreover, these cells displayed markedly enhanced clonogenicity and viability. Conversely, loss of function experiments, through genetic silencing of ALDH3A1, produced a marked perturbation of the redox proteome in tumor cells, evaluated through Western blot analysis for 4HNE adducts, and suppression of stem cell markers and stemness-related functions (tumorspheres, clonogenicity, viability). Thus, ALDH3A1 expression level clearly has a robust impact on these tumor cells functional state, favoring their potential tumorigenicity.

The ALDH3A subfamily, containing the dioxin-inducible ALDH3A1 and ALDH3A2 enzymes, is involved in the oxidation of medium- and long-chain aliphatic and aromatic aldehydes, and in the production of NADPH, suggesting that it may modulate redox-dependent signal transduction pathways involved in fate decision processes in tumors. However, diverse isoforms of ALDHs are able to detoxify aldehydes generated as byproducts of lipoperoxidation. ALDH3A1, the ALDH1A subfamily, comprising ALDH1A1, which synthesizes RA from retinaldehyde and, as such, is crucial in regulating RA signaling, has been used as a CSC marker for many solid tumors, including lung and melanoma [28,29]. In our study, in melanoma cells, we found no role for ALDH1A1 in cell survival and clonogenicity, and inhibitors of RA signaling failed to regulate PD-L1 expression in 3A1^high^ cells. Our data support the notion of a different specificity between the ALDHs in the regulation of tumor immune surveillance. 

In our tumor cell models, ALDH3A1 influences the redox state of cells, as demonstrated by the marked changes of the transcriptome profile, particularly of pro-inflammatory gene expression [30,31], including NFkB, COX-2, and mPGES1. Of note, we and others previously reported that PGE-2, the end product of the above enzymes, plays a key role in promoting EMT in tumor cells, demonstrating a close relationship between inflammation and EMT in tumor progression [32,33,34,35]. Here, we found that ALDH3A1 expression in tumor cells induces the overexpression of COX-2 and mPGES1, yielding high PGE-2 levels. Similarly, ALDH3A1 is associated with the overexpression of NFkB and EMT markers such as VIM, FN1, and Zeb1, revealing that the enzyme, through a direct effect on the redox state of tumor cell metabolism, is involved in the process that links stemness, EMT, and inflammation in tumor cells.

Inflammation and EMT share the property of bridging stemness and tumor immunity in tumor development [36]. Several studies have demonstrated that tumor-derived PGE-2 exerts a strong influence on immune evasion [21,22], as the prostanoid has been shown to influence the expression and release of immunosuppressive cytokines from tumor cells [21]. In this study, in ALDH3A1-silenced tumor cells, we observed a reduced expression of NFkB and COX-2, and inhibition of PGE-2 release, together with a marked subversion of the pattern of cytokine expression and release (decrease of immunosuppressive cytokines, such as IL-6 and IL-13, and increase of immunostimulating cytokines such as IL-12/23 and IFNγ), and the downregulation of the inhibitory signal PD-L1, indicating a switch to favor an active immune response. Analysis of PD-L1 by Western blot does not allow for the fine quantification of its expression in our tumor cell model; however, the extent of PD-L1 downregulation in 3A1^low^ cells supports the association between the two signaling molecules. Furthermore, while in overexpressing ALDH3A1 tumor cells the EMT markers Zeb1 and cMYC, known to be associated with PD-L1 expression [25,26], were upregulated, in ALDH3A1-ablated tumor cells they were downregulated. Consistently, in the attempt to link ALDH3A1 expression in tumor cells with systemic immune tone, we analyzed the activation of peripheral mononuclear blood cells (PBMC) against our tumor cellular models. The enhanced PBMC response, evaluated as cell proliferation, in coculture with ALDH3A1-ablated tumor cells, corroborated that the enzyme expression influences the tumor-intrinsic immune response. Taken together, our data indicate that the redox metabolism through the ALDH3A1 enzyme conveys signaling molecules originating from both inflammation and the EMT process into the control of the immune surveillance in tumor stem-cell-like cells, by regulating the expression of PD-L1 and the release of immunosuppressive cytokines. Thus, the ALDH3A1 enzyme, noted for its cytoprotective role in normal tissues, when overexpressed in cancer cells fosters the development of malignancy by activating a nearly seamless series of processes promoting stem cell development, cell differentiation, and immune suppressive cytokine release, which culminate with the enhancement of PD-L1 output and the ensuing evasion from the natural tumor immune constraint mechanism(s).

Although the activity of ALDHs is critical for stem cell development in several solid human tumors [11], confusion exists about its role in tumor progression. For example, Shao et al. posit that overexpression of ALDH enzymes, specifically ALDH1A3, is necessary but not sufficient to maintain the NSCLC stem cell population, and propose STAT3 activation as essential for cancer CSC function, hinting at the involvement of cytokines [37]. On the other hand, BRAF-mutated melanoma cells inoculated in mice or cultured in vitro have pointed to the role exerted by tumor-intrinsic factors on the immune system controlling tumor growth [21,38]. Among these factors, PGE-2and its synthetizing enzymes (COX-2 and mPGES1) assume a pivotal role since they consistently promote vigorous development across many tumor types. These reports, besides illustrating the mechanism whereby PGE-2 induces immunoevasion, provide evidence for a strategy that may lead to tumor eradication by combining PGE-2 inhibition and anti-PD-1 blockade. A discordant view, expressed in a commentary [39], claims that BRAF mutations in melanoma are not associated with PD-L1-induced immunoevasion, which, however, appears to be driven by IFNγ release in a subset of inflammatory melanoma patients.

Our work provides further insight on the molecular mechanism(s) underlying immunoevasion in lung and melanoma cancer cells, as we describe the concerted flow of signals that govern the PD-L1 expression. Following the initial ALDH3A1-provoked metabolic reprogramming and the CSC expansion, two synchronous processes, i.e., EMT and chemo-cytokines release, occur. The former induces marked phenotype changes, leading to the formation of cells endowed with metastatic traits, while the latter produces a variety of heterogeneous signals capable of enhancing PD-L1expression (cMYC, SOX2, cytokines). In light of the magnitude of the observed functional changes caused by ALDH3A1 overexpression in cancer cell lines, it appears plausible that the enzyme causes extensive cell redox metabolism reprogramming (for example, reduced 4HNE adducts), rather than being a mere marker of stemness. These findings highlight the complexity of the immunoevasion mechanism even in reductionist conditions such as the cultured cell lines employed here.

## 4. Materials and Methods

### 4.1. Cell Lines

WM266-4 (passages 5–20, ATCC^®^ CRL-1676™ certified by STRA, LGC Standards S.r.l., Milan, Italy) is a metastatic human melanoma cell line. WM266-4 ALDH3A1 knockdown (3A1^low^, passages 8–20), control cells (empty vector, Ctr), and WM266-4 overexpressing ALDH3A1 (3A1^high^) cells were cultured in DMEM 4500 high glucose (Euroclone) supplemented with 10% fetal bovine serum (FBS) (Hyclone, Celbio) and 2 mM glutamine, 100 units penicillin, and 0.1 mg/L streptomycin (Merck KGaA, Rome, Italy). HCC4006 (passages 5–20, ATCC^®^ CRL-2871™ certified by STRA, LGC Standards S.r.l.) is a lung adenocarcinoma derived from metastatic sites in pleural effusion. HCC4006 ALDH3A1 knockdown (3A1^low^, passages 8–20) and control cells were cultured in RPMI-1640 (Euroclone, Milan, Italy) supplemented with 10% FBS and 2 mM glutamine, 100 units penicillin and 0.1 mg/L streptomycin. Two different ALDH3A1- knockdown clones have been used. Lentivector plasmids for ALDH3A1 and control (Ctr) were obtained from Sigma Aldrich (St. Louis, MO, USA). All the plasmids were sequence verified. The sequence of plasmid inserted in cells clone 1 (Sh#1) is: 5′-CCGGGCTAAGAAATCCCGGGACTATCTCGAGATAGTCCCGGGATTTCTTAGCTTTTT-3′, the sequence of plasmid inserted in cells clone 2 (Sh#2) is: 5′- CCGGCGACAAGGTGATTAAGAAGATCTCGAGATCTTCTTAATCACCTTGTCGTTTTT-3′. To generate low-ALDH3A1(knockdown, Sh) cells, 1 × 10^6^ HEK293 cells (Life Technologies, Monza, Italy) were transfected with 2.25 μg of PAX2 packaging plasmid (Addgene, Teddington, UK), 0.75 μg of PMD2G envelope plasmid (Addgene), and 3 μg of pLKO.1 (Addgene) hairpin vector utilizing 12 μL of Lipofectamine 2000 (11668-019, Invitrogen, ThermoFisher, Monza, Italy) on 10-cm plates. Polyclonal populations of transduced cells were generated by infection with 1 MOI (multiplicity of infectious units) of lentiviral particles. At three days post-infection, cells were selected with 20 μg/mL neomycin/kanamycin (Sigma Aldrich) for one week [40]. To generate a stable knock-in, cells (overexpressing ALDH3A1, 3A1^high^) were seeded on six-multiplates and transfected with lentiviral particles containing nucleotide sequences encoding for ALDH3A1 (Origene RC202440L1V, Lenti ORF particles, ALDH3A1 (Myc-DDK tagged)-Human. High 3A1 cells were generated by G418 selection for 10 days, selecting at least two clones for each cell line. Immortalized human keratinocytes HaCaT cells (passages 3–7) were acquired from Voden Medical (Meda, Italy). All the cell lines were certified by STRA and cultured as recommended. All the cell lines were immediately expanded after delivery (up to 6 × 10^7^ cells) and frozen down (1 × 10^6^/vial) such that all the cell lines could be restarted after a maximum of 10 passages every three months from a frozen vial of the same batch of cells. Control of mycoplasma was done from frozen vials. Melanoma cell lines were established in vitro from surgically resectable tumor lesions of cutaneous metastatic melanoma patients #4478D (kindly provided by Prof. Michele Maio, University of Siena, Italy). These melanoma cell lines were cultured in vitro with RPMI 1640 supplemented with 10% FBS, 20 mM HEPES, penicillin (200 U/mL), streptomycin (200 ug/mL), and 2 mM glutamine. Peripheral blood mononuclear cells (PBMC, ATCC PCS-800-011™ certified by STRA, LGC Standards S.R.L., Sesto San Giovanni-Milan, Italy) are a heterogeneous population of blood cells with a single round nucleus and include macrophages, dendritic cells, monocytes, and lymphocytes. They have a limited lifespan in culture and should only be thawed immediately prior to their intended use. These cells were cultured in RPMI 1640 supplemented with 10% FBS, 20 mM HEPES, penicillin (200 U/mL), streptomycin (200 ug/mL), and 2 mM glutamine.

### 4.2. Immunoblot Analysis

Where indicated, cells were treated with RAR antagonist AGN193109 (5758, Tocris Bioscience, Bristol, UK) and RXR antagonist UVI3003 (3303, Tocris Bioscience) for 48 h (1 µM).

Total protein lysates were obtained using celLytic MT (C2978-50 mL, Sigma Aldrich), a cell lysis reagent, as described [32]. Antibodies used are as follows: anti-Oct-4A (2840 Cell Signaling, Milan, Italy), anti-Sox2 (3579 Cell Signaling), anti-KLF4 (4038 Cell Signaling), anti-Nanog (4903 Cell Signaling), anti-c-MYC (5605 Cell Signaling), anti-PD-L1 (13684; Cell Signaling); anti-ALDH3A1 (TA332730, Origene); anti-mPGES1 (160140) and anti-COX-2 (160112, Cayman Chemical, Arcore, Italy); anti-CD133 (PA1217, Boster), anti-NFkB (sc-372, Santa Cruz, CA, USA), and anti-β-actin (Sigma Aldrich). Images were digitalized with CHEMI DOC Quantity One program (Biorad, Milan, Italy

### 4.3. MTT Assay

Cell proliferation was quantified by Vybrant MTT (M5655-1G, Sigma Aldrich) cell proliferation assay as described [33]. Briefly, cells (3 × 10^3^) were seeded in 96-multiwell plates in medium with 10% serum for 24 h and then grown for 48 h in a complete medium with 0.1 or 10% FBS in the presence or not of CM037 (10 μM, ChemDiv Inc., San Diego, CA, USA) every 24 h. Data are reported as cell viability at 540 nm absorbance/well. 

### 4.4. Clonogenic Assay

For clonogenic assay, WM and HCC cells were cultured for 48 h in a complete medium with 10% FBS or treated with CM037 (10 µM every 24 h). Cells were plated in 60-mm culture dishes (at a density of 1000 cells/dish) in a medium containing 1% FBS, and then kept in a humidified incubator at 37 °C and 5% CO_2_ for two weeks. Colonies (>75 cells) with 50% plate efficiency were fixed and stained with 0.05% crystal violet (C3886-25G, Sigma Aldrich) in 10% ethanol and counted. Data are expressed as the number of colonies that arise after treatment of cells [33].

### 4.5. Analysis of ALDH3A1-Catalyzed Dehydrogenase Activities from Cell Lysates

Briefly, cells (WM, HCC, and MEL) were washed with ice-cold PBS to remove residual medium. Then 400 μL of celLytic MT buffer (Sigma Aldrich) containing 1 mM PMSF (000000010837091001, Sigma Aldrich) was added to each 10-cm dish. Plates were incubated on ice for 5 min and scraped, and lysates were collected. Lysates were centrifuged for 10 min at 16,000× *g* in a microcentrifuge at 4 °C. Protein concentrations were measured using the Bradford reagent (B6916, Sigma Aldrich). Then, 300 μg of cell lysate was used in the activity assay. ALDH3A1 activity in cell lysates was measured in 100 mM Na_2_HPO_4_ buffer (S5136, Sigma Aldrich) at pH 9, with 10 mM NADP+ (000000010128031001, Sigma Aldrich) and 10 mM benzaldehyde (B1334, Sigma Aldrich). The lysate was used to detect ALDH activity at 25 °C by monitoring NAD(P)H formation from NADP(+), at 340 nm in a spectrophotometer Infinite F200 Pro (Tecan Life Sciences, Mannedorf, Switzerland). The assay mixture (0.8 mL) contained 100 mM sodium pyrophosphate pH 9.0, 10 mM NADP+, and 300 µg of sample protein. The reaction was started by adding benzaldehyde (10 mM) to the cuvette. The assay including the controls contained 1% (v/v) DMSO (D8418, Sigma Aldrich); the compound was tested at 10 μM to monitor the extent of ALDH inhibition in cell lysates [33,41]. Enzyme-specific activity was expressed as % of nmol NADPH/minute/mg protein.

### 4.6. Prostaglandin E2 Express ELISA Assay

PGE2 was measured by an EIA kit (#500141, Prostaglandin E2 Express ELISA kit-Monoclonal, Cayman Chemical). Cells were exposed to 1% for 48 h and treated for 24 h with 10 µM arachidonic acid (10931, Sigma Aldrich). Cell culture supernatants were assayed directly following the manufacturer’s instructions. PGE2 concentration was expressed as (pg/mL), normalized to total protein concentration.

### 4.7. ELISA Immuno-Assay

IL-6 (#D6050), IL-12/IL-23 p40 (#D400), IL-13 (#D1300) and IFNγ (#DIF50) were determined in supernatants using a Quantikine kit (R&D System, Milan, Italy). 3 × 10^4^ cells were exposed to 1% for 48 h. The conditioned media were collected, diluted in the standard diluents, and assayed as indicated in the manufacturer’s instructions. Data were reported as cytokines levels (pg/mL), normalized to total protein concentration.

### 4.8. Real-Time PCR

Total RNA was prepared using a RNeasy Plus Kit (#74134 Qiagen, Milan, Italy) following the manufacturer’s instructions. One microgram of RNA was reverse-transcribed using QuantiTect Reverse Transcription Kit (#205313 Qiagen), and quantitative RT-PCR (QPCR) was performed using QuantiNova SYBR Green PCR Kit (#208056 Qiagen) in a Rotor-Gene Q PCR machine (Qiagen). Fold change expression was determined by the comparative Ct method (ΔΔCt) normalized to 60 S Ribosomal protein L19 expression. qRT-PCR data are represented as fold increase relative to nontreated cells (Control), which was set to 1. Primers for quantitative QPCR are listed in Appendix A. ALDH3A1 primers were from Qiagen (PPH07009A). 

### 4.9. Immunofluorescence Analysis

The localization of transcription regulator NFkB was monitored by confocal analysis. A total of 3 × 10^4^ WM were seeded on 1-cm circular cover glass coverslips placed in the bottom of a 12-well multiplate. After 48 h WM were fixed in cold acetone (5 min), washed in PBS, and incubated with 3% BSA (45 min). Cells were then incubated for 16 h with anti-NFkB (1:50) (sc-372, Santa Cruz, CA, USA) antibody in PBS containing 0.5% BSA. After incubation (1 h) with the secondary antibody anti-Rabbit IgG, Alexafluor 568 (A-11011, ThermoFisher Scientific), samples were washed with PBS and treated with DAPI in PBS (1:5000). After incubation, cells were washed, the coverslips were mounted with Fluoromount Aqueous Mounting Medium (F4680, Sigma-Aldrich), and images were taken using confocal microscope (Leica SP5, Buccinasco, Milan-Italy) at 40× magnification.

### 4.10. Tumorsphere Formation In Vitro

This assay tests the ability of single cells to form tumorspheres, the in vitro surrogate of stem-like cells [42]. Cells (2 × 10^5^ cells/well in 1.5 mL of medium) were distributed into an ultralow-attachment six-well plate. All tumorspheres were grown in DME-F12 medium (Gibco, Milan, Italy), supplemented with penicillin/streptomycin, L-glutamine, B27 supplement (1×, #17504-044, Life Technologies), bFGF (20 ng/mL, #13256029, Gibco), and hEGF (20 ng/mL, #10605-HNAE, Gibco), and allowed to grow for 7–10 days, or until the majority of spheres reached a diameter of 60 μm. Tumorspheres were treated with CM037 (10 μM) every 24 h. Tumorspheres were counted and then harvested followed by protein extraction or split for second and third tumorsphere generation and lysed for protein extraction [33].

### 4.11. Human Cytokine ELISA Plate Array

We used the Human Cytokine ELISA Plate Array (#EA-4001, Signosis Inc., Santa Clara, CA, USA), a fast and sensitive assay used for the quantitative comparison of 32 cytokines among different samples. Cells were exposed to a medium with 1% serum for 48 h in the presence/absence of CM037 (10 μM). The cell culture supernatants from each sample were incubated with the cytokine ELISA plate, and the captured cytokine proteins were subsequently detected with a cocktail of biotinylated detection antibodies. The test samples were allowed to react with a pair of antibodies, resulting in the cytokines being sandwiched between the solid phase and enzyme-linked antibodies. After incubation, the wells were washed to remove unbound, labeled antibodies. The plate was further detected with HRP luminescent substrate. The level of expression for each specific cytokine is directly proportional to the luminescent intensity. Data are reported as % of fold change vs. Ctr cells. 

### 4.12. Conditioned Media from Tumor Cells

Tumor cells were seeded in a 24-well plate at 5 × 10^4^ cells/well in 1 mL of complete medium. Cells were exposed to 10% FBS for 48 h. The cell culture supernatants from each sample were harvested and maintained at −80 °C.

### 4.13. CFSE Cell Proliferation Assay

To monitor distinct generations of proliferating dyed PBMC cells, we used the CellTrace™ CFSE kit (# C34554, Thermo Scientific) according to the manufacturer instructions. Alive cells are covalently labeled with a very bright, stable dye. Every generation of cells appears as a different peak on a flow cytometry histogram. Briefly, 2 × 10^7^ cells were centrifuged for 5 min at 300× *g*, re-suspended in 10 mL of CellTrace™ CFSE staining solution (5 µM), and incubated for 20 min in a 37 °C water bath. Then cells were centrifuged for 5 min at 300× *g* and the pellet was resuspended in prewarmed OpTmizer™ T Cell Expansion SFM. We reserved 1 mL of cells for unstained control and 1 mL of cells for a stained but unstimulated control. Aliquots of stained cells were distributed into 24-well plates. Two different experimental conditions were set up: PBMC incubated with conditioned media (CM) from tumor cells or tumor cells (in adhesion) co-cultured with PBMC (in suspension). We assessed PBMC (without activator) in a co-culture with tumor cells or PBMC with activators incubated with CM. PBMCs were activated with human T-Activator anti-human CD3 antibody (2 μg/mL, 317302, BioLegend, Koblenz, Germany), oranti-CD28 antibody (clone 15E8, CBL517, Sigma Aldrich), or IL-2 2U (57600-1VL, Sigma Aldrich). Every 24 h up to 144 h, the CFSE fluorescence was analyzed using a Guava easyCyte Single Sample Flow Cytometer at 488 nm (Merck Millipore, Milan, Italy).

### 4.14. Tumor Sample Analysis

Archival paraffin-embedded tissue samples from patients with pulmonary adenocarcinoma or skin melanoma were collected from the University Hospital of Siena and written informed consent to perform this analysis was obtained from all patients. Representative formalin-fixed, paraffin-embedded tumor tissue blocks were selected and 4-µm sections for each lesion were prepared for immunohistochemical analysis. Antigen retrieval was performed for 20 min in citrate buffer (pH 6.6) in a microwave at 500 W for ALDH3A1 and COX-2. The sections were then allowed to cool down to room temperature for 20 min. After inactivating endogenous peroxidase activity and blocking cross-reactivity with 3% BSA, the slides were incubated at 4 °C for 18 h with a dilute solution of anti ALDH3A1 (1:75, #MABC121, Merck Millipore), COX-2 (1:250, #12282, Cell Signaling) antibodies. The location of primary antibodies was determined by subsequent application of biotin-conjugated anti-primary antibody, streptavidin–peroxidase, and diaminobenzidine (#04293, Sigma Aldrich). The staining was developed using a commercial immunoperoxidase staining kit following the manufacturer’s instructions (biotin–streptavidin complex method, #20774, Merck Millipore). The slides were counterstained with hematoxylin (#HHS16, Sigma Aldrich). In addition, all samples were stained withPD-L1 antibody SP263 (Ventana/Roche, Milan, Italy, USA) on an automated platform (Benchmark, Ventana/Roche, USA) following the manufacturer’s protocol. Two independent investigators (C.B. and S.A.) scored each of the sections without knowledge of the histologic diagnosis or staining pattern of the other marker. The assessment of expression levels of ALDH3A1 and COX-2 included the staining intensity and the percentage of stained cells. The staining intensity was scored as 0 = no staining, 1 = faint, 2 = moderate, and 3 = strong expression; the results were categorized according to the following distribution: 0 ≤ 9%, 1 = 10–50%, 2 ≥ 51% staining. The ALDH3A1 expression score was determined as a combined score of staining intensity and distribution. Samples with a final immunoscore ≥2 were considered ALDH3A1-positive. For PD-L1 detection, nuclei and membrane expression, respectively, and the labeling index was calculated as the percentage of labeled nuclei/membrane of the total number of cells that were counted. There was a high correlation between the two scores, and in the few discrepant cases a consensus was reached after joint review. Antibody binding was microscopically recognizable as brown cytoplasmic staining.

### 4.15. Invasion Assay

Cell invasion was performed by the Boyden chamber technique (Neuroprobe 48-well microchemotaxis chamber) (BiomapSnc, Agrate B.za, MI, Italy), with a filter coated with gelatin (Sigma Aldrich) [43]. First 1.25 × 10^4^ cells (WM Ctr, 3A1^low^, 3A1^high^ and HCC Ctr, 3A1^low^) were added to the upper wells of the chamber. Lower wells contained 5% FBS as a chemoattractant. After 18 h of incubation, cells were fixed and stained with a Diff-Quik kit (Biomap Snc, Agrat B.za, MI, Italy). Migrated cells present in five fields/well were counted at 40× original magnification. Data are reported as the number of counted cells/well.

### 4.16. Statistical Analysis

Results are expressed as means ±SD or ±SEM. Statistical analysis was carried out using Student’s *t*-test and two-way ANOVA, followed by a Bonferroni post-test for multiple comparison. *p* < 0.05 was considered statistically significant.

## 5. Conclusions

Despite the clinical success of antibodies against the immune checkpoint regulators such as PD-L1/PD-1, only a subset of people shows durable responses. Three basic immune profiles that correlate with a patient’s response to anti-PD-L1/PD-1 therapy have been identified: the inflamed tumor, the immune-excluded tumor, and immune desert tumors [31]. In all profiles, the intrinsic tumorigenicity of tumor cells and the state of activated T cell immunity or tolerance play a pivotal role [31]. Of note, also in the first profile, the immune-inflamed phenotype, the response to anti PD-1/PD-L1 treatment is not assured, indicating that immune cell infiltration is necessary but insufficient for inducing a response, as other elements, potentially intrinsic to tumor cells, have to be considered. In this context, the tumor cell redox metabolism has an important role in T cell dysfunction and the response to immunotherapy [44]. The role of immune cell recruitment in shaping the response to immunotherapy (i.e., PD-L1 blockade) remains somewhat controversial, as evidence has indicated that the prevailing mechanism is intrinsic to the tumor through the activation of EMT [17,25], while other recent reports have speculated on the essential role of the adaptive immune system [45,46].

In this study we show that tumor cells overexpressing ALDH3A1 by the co-option of inflammatory and EMT signaling molecules are fully capable of mounting, in an autonomous manner, an adequate immune response. Ablation of ALDH3A1 expression and activity appears to be a new strategy for targeting the tumor immunosuppressive network.

## Figures and Tables

**Figure 1 cancers-11-01963-f001:**
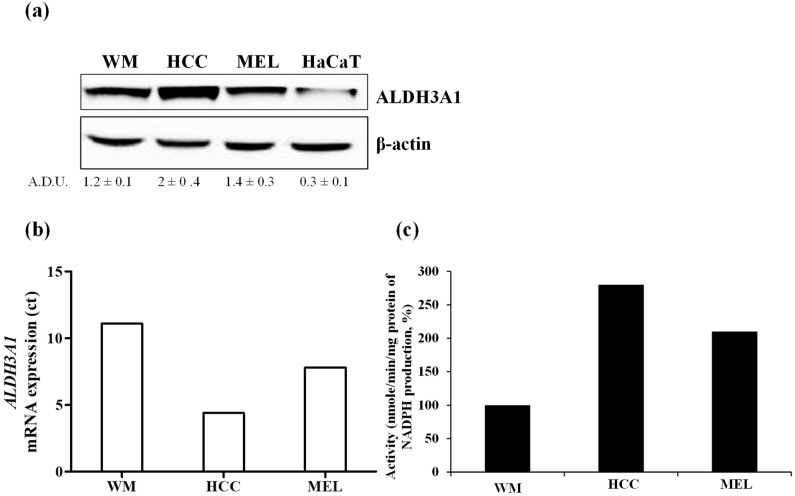
ALDH3A1 expression and activity in metastatic melanoma and NSCLC cells. (**a**) Protein expression of ALDH3A1 in metastatic melanoma cells (WM), NSCLC (HCC), and metastatic melanoma cells from patients BRAF WT (MEL) and keratinocytes (HaCaT) cultured for 48 h in complete medium (DMEM 4500 mg/L glucose for WM and HaCaT; RPMI1640 for HCC and MEL) with 10% FBS. β-actin has been used to normalize loading. (**b**) mRNA expression of ALDH3A1 in WM, HCC, or MEL cultured for 36 h in a complete medium with 10% FBS. Data expressed as Ct. (**c**) ALDH3A1 activity in tumor cells cultured for 24 h in complete medium with 10% FBS. Data expressed as % of nmol NADPH/minute/mg protein.

**Figure 2 cancers-11-01963-f002:**
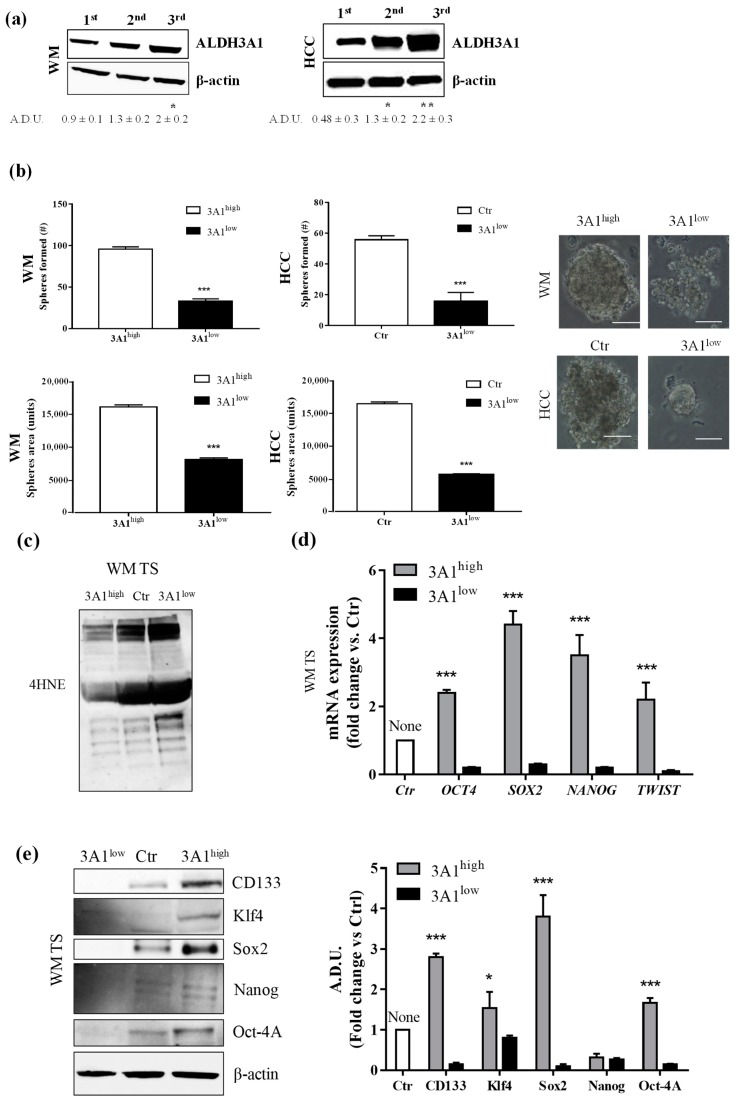
ALDH3A1 expression controls the stemness phenotype. (**a**) ALDH3A1 protein expression in lysates from spheres from first to third generation from WM (left) or HCC (right) maintained in basal condition. β-actin has been used to normalize loading. Arbitrary densitometry unit (A.D.U.) ± SD was reported. * *p* < 0.05, ** *p* < 0.01 vs. 1st generation of tumorspheres. (**b**) Representative images of third-generation spheres from WM (top) or HCC (bottom), high and Ctr (left) or low (right) 3A1. Sphere number (top) and sphere area (bottom) from third-generation spheres in WM3A1^high^ and HCC Ctr or 3A1^low^. *** *p* < 0.001, vs. 3A1^high^ cells. (**c**) Accumulation of 4-HNE adducts in WM tumorspheres expressing different level of ALDH3A1. (**d**) mRNA expression of *TWIST*, *SOX2*, *OCT4* and *NANOG* in third-generation spheres (TS) from WM 3A1^high^ or 3A1^low^. *** *p* < 0.001, vs. 3A1^low^ cells. (**e**) Protein expression of CD133, Klf4, Sox2, Oct4 and Nanog in tumorspheres (TS) from WM 3A1^high^ or 3A1^low^. β-actin used to normalize loading. * *p* < 0.05 and *** *p* < 0.001, vs. 3A1^low^ cells. (**f**) Accumulation of 4-HNE adducts in HCC tumorspheres expressing different level of ALDH3A1. (**g**) mRNA expression of *TWIST*, *SOX2*, *OCT4* and *NANOG* in third-generation spheres (TS) from HCC 3A1^high^ or 3A1^low^. *** *p* < 0.001, vs. Ctr cells. (**h**) Protein expression of CD133, Klf4, Sox2, Oct4, and Nanog in tumorspheres (TS) from HCC Ctr or 3A1^low^. β-actin used to normalize loading. *** *p* < 0.001, vs. Ctr cells.

**Figure 3 cancers-11-01963-f003:**
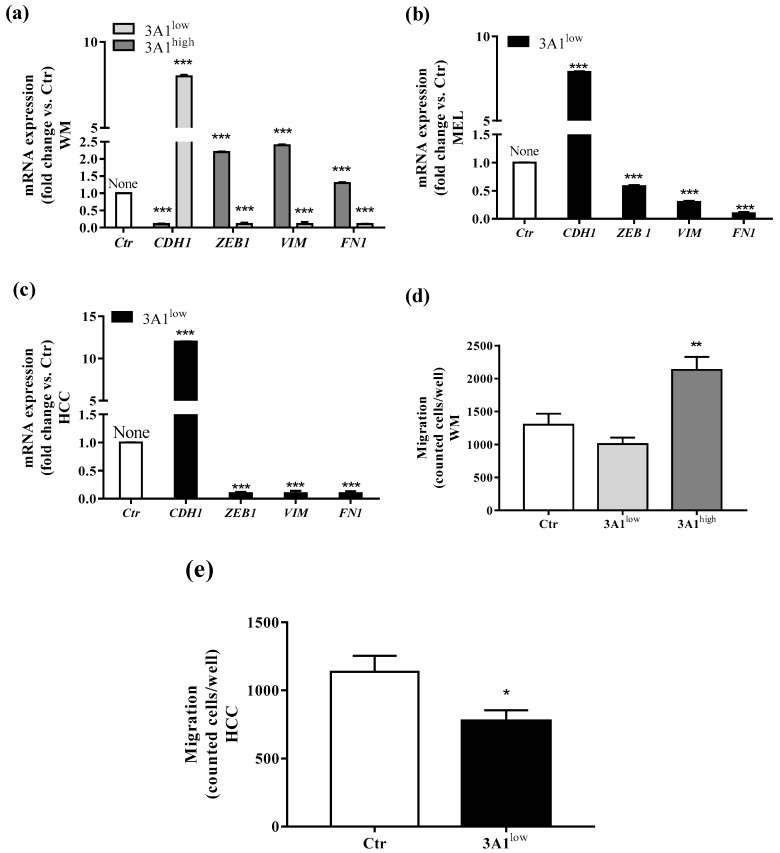
ALDH3A1 expression controls EMT markers. (**a**) mRNA expression of *CDH1*, *Zeb1*, *VIM*, and *FN1* in WM3A1^low^ or 3A1^high^ cells. (**b**) mRNA expression of *CDH1*, *Zeb1*, *VIM*, and *FN1* in MEL3A1^low^ cells. (**c**) mRNA expression of *CDH1*, *Zeb1*, *VIM*, and *FN1* in HCC 3A1^low^ cells. All cells were maintained for 48 h in a medium with 10% FBS. Data are reported as fold change vs. Ctr cells. *** *p* < 0.001 vs. Ctr cells. (**d**) WM migration through a gelatin-coated filter toward serum gradient. Data are reported as number of cells counted/well. (*n* = 3). ** *p* < 0.01 vs. WM 3A1^low^. (**e**) HCC migration through a gelatin-coated filter toward serum gradient. Data are reported as number of cells counted/well. (*n* = 3). * *p* < 0.5 vs. HCC Ctr.

**Figure 4 cancers-11-01963-f004:**
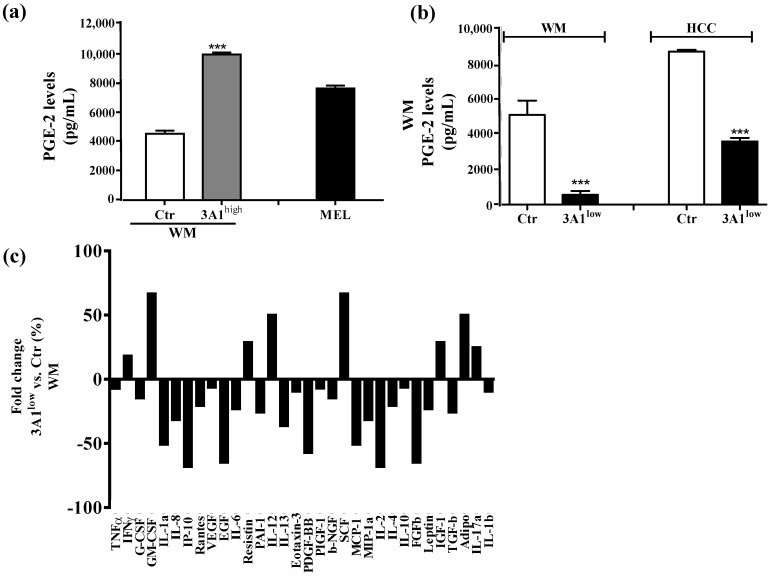
ALDH3A1 controls PGE-2 signaling and affects the release of cytokines important in immune surveillance. (**a**) PGE-2 levels (pg/mL) in WM Ctr and 3A1^high^ and in MEL. (**b**) PGE-2 levels (pg/mL) in WM and HCC Ctr and 3A1^low^. All cells were maintained in 1% FBS for 48 h (arachidonic acid, 10 µM, as substrate added in the last 24 h). *** *p* < 0.001 vs. Ctr cells. (**c**) Human cytokine ELISA Array in chemiluminescence in WM 3A1^low^ cells. Data are expressed as percentage of fold change vs. Ctr cells. (**d**) Western blot analysis (left panel) and immunofluorescence analysis (right panels) for NFkB p65 expression and localization in 3A1^low^ and 3A1^high^ WM cells maintained in 10% FBS for 24 h. Arrows indicate the nuclear localization of NFkB p65. (scale bar: 120 μm for 40× panels, 100 μm for 63× panels and 50 μm for 63× (zoom 5×) panels).

**Figure 5 cancers-11-01963-f005:**
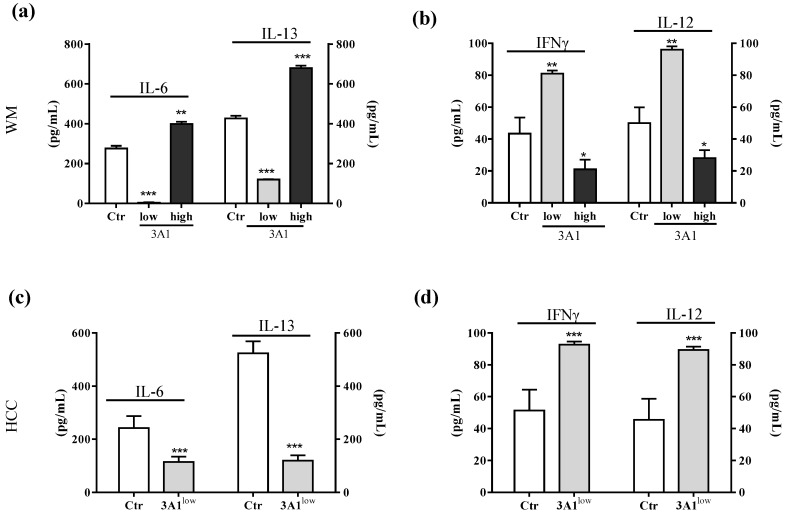
ALDH3A1 modulation reduces IL-6 and IL-13 release and increases IL-12 and IFNγ levels. (**a**) ELISA for IL-6 and IL-13 in WMCtr, 3A1^low^ or 3A1^high^. (**b**) ELISA for IFNγ and IL-12 in WMCtr, 3A1^low^ or 3A1^high^. (**c**). ELISA for IL-6 and IL-13 in HCC Ctr or 3A1^low^. (**d**) ELISA for IFNγ and IL-12 in HCC Ctr or 3A1^low^. (**e**) ELISA for IL-6 and IL-13 in MEL. (**f**) ELISA for IFNγ and IL-12 in MEL. Data are expressed as pg/mL. All cells are maintained for 48 h in 1% FBS. *** *p* < 0.001, ** *p* < 0.01, * *p* < 0.05 vs. Ctr.

**Figure 6 cancers-11-01963-f006:**
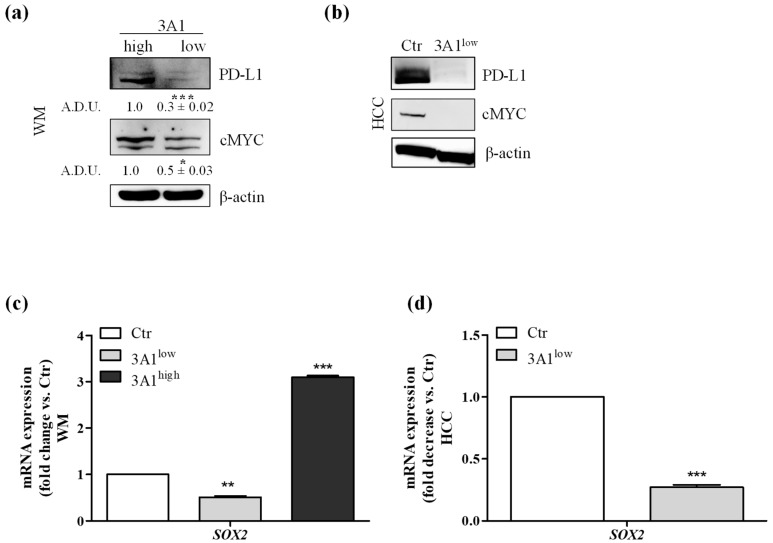
Modulation decreases the expression of immunomodulatory proteins. (**a**) PD-L1 and cMYC protein expression analyzed by Western blot in WM, 3A1^high^ or 3A1^low^, * *p* < 0.05 and *** *p* < 0.001 vs. 3A1^high^ cells. (**b**) PD-L1 and cMYC protein expression in HCC Ctr or 3A1^low^. All cells were maintained in 10% FBS for 48 h. β-actin has been used to normalize loading. (**c**) Sox-2 mRNA expression in WM (Ctr, 3A1^low^ or 3A1^high^). (**d**) Sox-2mRNA expression in HCC (Ctr or 3A1^low^#1). Cells were maintained in 10% FBS for 48 h. Data are reported as fold change/decrease vs. Ctr *** *p* < 0.001, ** *p* < 0.01 vs. Ctr.

**Figure 7 cancers-11-01963-f007:**
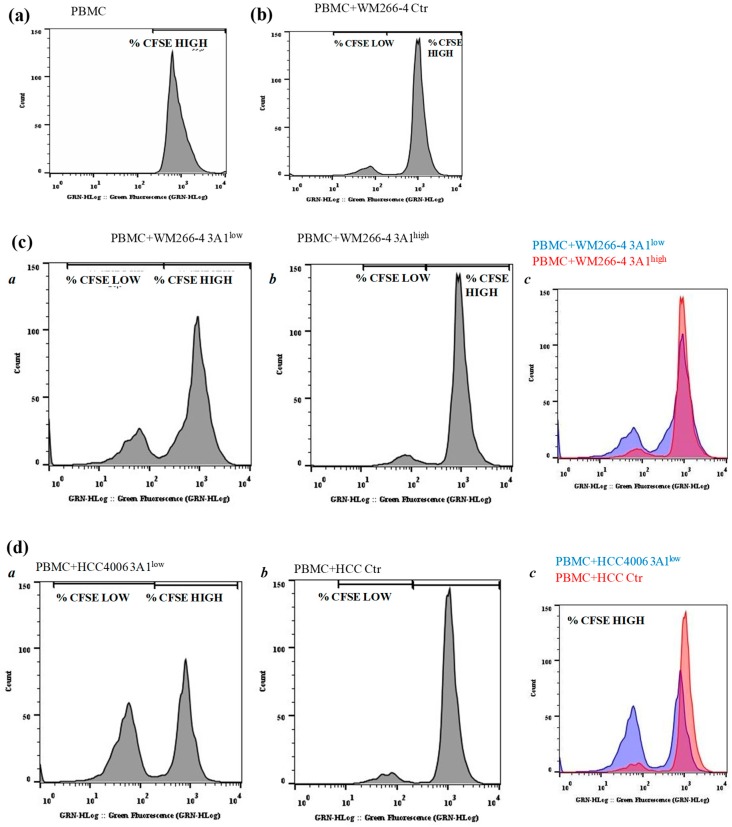
ALDH3A1 expression in human melanoma decreases PBMC proliferation. (**a**–**d**) CFSE-labelled PBMCs were kept alone (**a**) or incubated for five days with WM Ctr (**b**) WM 3A1^low^ and 3A1^high^ (**c**) or HCC Ctr, or 3A1^low^ (**d**) (panels a and b, and c for merged) harvested every day and assayed for proliferation using flow cytometry. Representative histogram plots of live-gated PBMCs undergoing proliferation measured by CFSE are shown. % of CFSE high and low cells are indicated.

**Figure 8 cancers-11-01963-f008:**
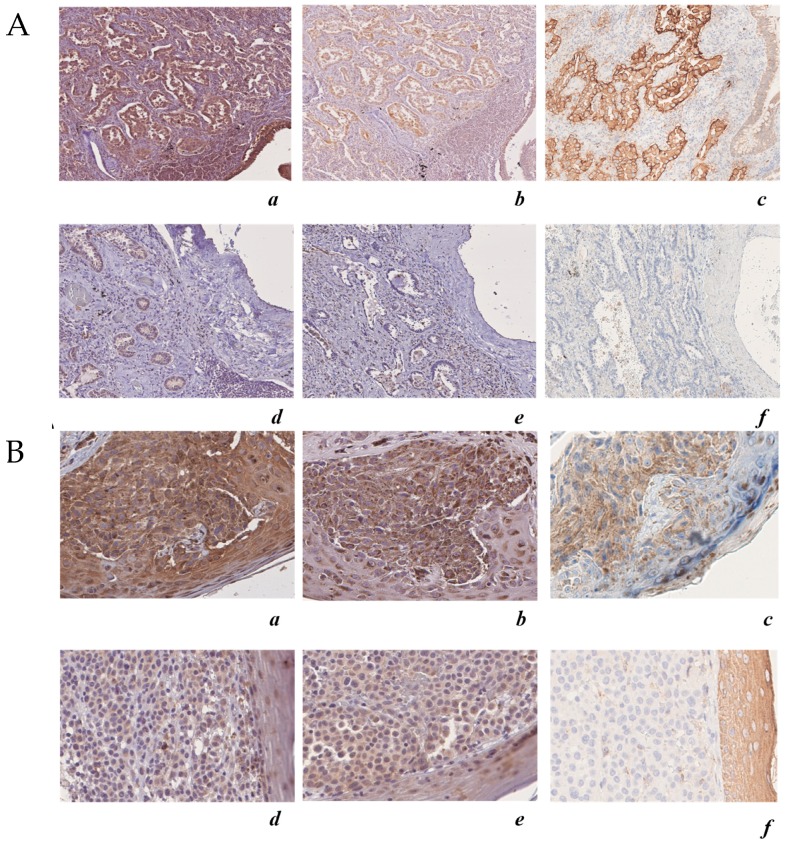
ALDH3A1 expression in human melanoma and lung adenocarcinoma is associated with COX-2 and PD-L1 expression. (**A**) Upper panels: case 1 lung adenocarcinoma, immunohistochemical analysis (200×) of: (**a**) ALDH3A1 (high expression) (200×); (**b**) COX-2 (high expression); (**c**) PD-L1 (high expression). Lower panels: case 2 lung adenocarcinoma immunohistochemical analysis (200×) of: (**d**) ALDH3A1 (low expression); (**e**) COX-2 (low expression); (**f**) PD-L1). (**B**) Upper panels: case 1 skin melanoma, immunohistochemical analysis (200×) of: (**a**) ALDH3A1 (high expression); (**b**) COX-2 (high expression); (**c**) PD-L1 (high expression). Lower panels: case 2 skin melanoma immunohistochemical analysis (200×) of (**d**) ALDH3A1 (high expression); (**e**) COX-2 (high expression); (**f**) PD-L1 (high expression). The hematoxylin–eosin-stained sections are included as a basal control. Scale bars indicate 100 μm, magnification 4× and 10×.

**Table 1 cancers-11-01963-t001:** Proliferation of CFSE-stained PBMC cultured in conditioned media from tumor cells.

Cells	MFI	CFSE % ^LOW^	CFSE % ^HIGH^
PBMC	929	-	100
PBMC + L-2 + WM266-4 Ctr	546	11.8	88.2
PBMC + IL-2 + WM266-4 ALDH3A1^high^	653	17.6	82.4
PBMC + IL-2 + WM266-4 ALDH3A1^low^	402	32.2	67.7
PBMC + WM266-4 Ctr	627	9.85	90.15
PBMC + WM266-4 ALDH3A1^high^	745	9.04	90.96
PBMC + WM266-4 ALDH3A1^low^	125	24.8	75.2
PBMC + IL-2 + HCC4006 Ctr	769	16.6	83.4
PBMC + IL-2 + HCC4006 ALDH3A1^low^	306	54.4	45.6
PBMC + HCC4006 Ctr	838	9.47	90.53
PBMC + HCC4006 ALDH3A1^low^	442	48.7	51.3

Mean fluorescence unit (MFI) and % of CFSE low (proliferating) and high (not proliferating) cells are reported.

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
