# Peer review of "ALDH3A1 Overexpression in Melanoma and Lung Tumors Drives Cancer Stem Cell Expansion, Impairing Immune Surveillance through Enhanced PD-L1 Output"

_cancers, 2019, doi:10.3390/cancers11121963_

Round 1

Reviewer 1 Report

The authors addressed two cancer types that respond well to immune check inhibitors (PD-1..). They demonstrate a clear correlation between ALH3A1 and PD-L1 and COX-2 in both cancer types. Their finding have a significant translation value and could be essential for understanding the connection between stemness, antitumor immunity and inflammation.This aspect could be discussed further, particularly at the end of the discussion.

Reviewer 2 Report

The manuscript by Terzuoli et al. demonstrated that the overexpression of ALDH3A1 is highly correlated to the cancer stemness and immune surveillance in melanoma and lung cancer. Authors proved the association of ALDH3A1 and cancer stemness via the enforced ALDH3A1 in Melanoma cells WM266-4. Moreover, the association between ALDH3A1 and malignancy was validated in clinical samples. This issue is interesting but there are still some concerns about this manuscript.

Major concerns:

The expression of cancer stemness markers in HCC TS 3A1 (high) cells? The data is missing and cannot be found in Figure 2e and 2f. What is the effect of high ALDH3A1 on the EMT markers in MEL and HCC cells? Discussion section is too long and confused. It is suggested to be shortened and better organized.

Minor concern

Some figures presented in this manuscript are not good and difficult to be understood. Authors are supposed to figure out a better way to present their data for easy reading. What is the meaning of EV in Figure 2b, MW cells? Although the definition of EV was shown in Supplementary Materials. Please separate the protein and gene, or mark them clearly. For example, line 132. E-cadherin (encoded by CDH1); line 133, fibronectin (encoded by FN1), vimentin (encoded by VIM). In Figure S3g, the NF-kB p65 staining showed the differently bright in 3A1 low. The red color in 3A1 low P65/DAPI merge image (low right) showed much darker p65 alone image (low middle). Western blot showed ALDH3A1 highly enhanced the p65 expression. However, immunofluorescence staining showed that most p65 located in cytosol, not nuclei. It was unable to demonstrate that the p65 activity was increased by ALDH3A1. It is suggested to demonstrate its activation by phospho-p65 antibody. In line 165, there is a type error in NSCLC. It is difficult to compare the difference among IL-6, IL-13, IL-12, and IFNg. Please combine Figure 5a/5e and 5b/5f. In figure S5a, what is control lysate? Why actin is missing in control lysate? In table S3 and S4, the differentiation rate is from PBMC or T cells? Please move the quantification of PBMC differentiation into the Figure 7 for easy reading. Actin is b-actin? It is suggested to correct it.

Reviewer 3 Report

The work by Terzuoli et al explores the role of ALDH1A3 in several cellular activities related to stemness, such as tumorsphere formation and pluripotency adn EMT genes expression, and immunoevasion. Interestingly, authors show patient data indcitating a correlation between the expression of the enzyme with COX-2 and PD-L1, further supporting a role of ALDH1A3 in tumor immunity.

Although potentially interesting mainly due to their observations related to immune surveillance, I find this work would need improvement in several aspects:

Major points

Data for ALDH1A3low, high and control cells must be shown in every figure. As they are now, it is difficult to understand why in some figures data for high and low plus the control situation are included, while in others only data for one or two of these conditions are shown. In fact, sometimes the figure legend estates there ara data for both high and low, while the figures contains only data o¡for one of the situations (figure 2e-f as an example). This fact, together with the lack of consistency in showing results for the 3 cells lines used in the study (lack of MEL in figure 2, for example), makes the figures difficult to interpret and evaluate. Although the link between stemness, EMT and immune suppression is clear from a theoretical point of view, data presented here are merely correlative: no morphological or functional assays demonstrating that EMT genes expression have cellular consequences in shown (changes in cellular shape, cytoskeleton reorganization, changes in migration/invasion). Figure 6 linking PD-L1, MYC and SOX2 expression with ALDH1A3 needs to be completed showing the 3 markers for the 3 cells lines in control, high and low conditions. Flow cytometry staining for PD-L1 is the standard way of quantifying membrane expression of this marker. Figure 7: what is the effect on PBMCs of the control (non-infected) cells?

Minor points:

Why showing wild type MEL cells in figure 5? Including some kind of labelling close to the histology (figure 8) micrographs would help the readers Quality of some of the WB images could be improved Data for NF-kB are imprtant enough to be shown in main figures

Round 2

Reviewer 3 Report

no further comments